# Analysis of Tissue-Specific Defense Responses to *Sclerotinia sclerotiorum* in *Brassica napus*

**DOI:** 10.3390/plants11152001

**Published:** 2022-07-31

**Authors:** Jie Liu, Rong Zuo, Yizhou He, Cong Zhou, Lingli Yang, Rafaqat Ali Gill, Zetao Bai, Xiong Zhang, Yueying Liu, Xiaohui Cheng, Junyan Huang

**Affiliations:** 1Key Laboratory of Biology and Genetic Improvement of Oil Crops, Ministry of Agriculture and Rural Affairs of the PRC, Oil Crops Research Institute, Chinese Academy of Agricultural Sciences, Wuhan 430062, China; whjiejiel@163.com (J.L.); hu086zr@163.com (R.Z.); yizhhe@163.com (Y.H.); zhoucong@caas.cn (C.Z.); yanglingli1120@163.com (L.Y.); drragill@caas.cn (R.A.G.); baizetao_2005@163.com (Z.B.); hbzhangxiong@126.com (X.Z.); lyy680608@126.com (Y.L.); 2Center of Integrative Biology, Interdisciplinary Science Research Institute, College of Life Science and Technology, Huazhong Agricultural University, Wuhan 430070, China

**Keywords:** *Brassica napus*, Sclerotinia stem rot, transcriptome, tissue-specific, defense response

## Abstract

Sclerotinia stem rot (SSR) caused by *Sclerotinia sclerotiorum* (*S. sclerotiorum*) is the main disease threat of oilseed rape (*Brassica napus*), resulting in huge economic losses every year. SSR resistance manifests as quantitative disease resistance (QDR), and no gene with complete SSR resistance has been cloned or reported so far. Transcriptome analysis has revealed a large number of defense-related genes and response processes. However, the similarities and differences in the defense responses of different tissues are rarely reported. In this study, we analyzed the similarities and differences of different tissues in response to *S. sclerotiorum* at 24 h post inoculation (hpi) by using the published transcriptome data for respective leaf and stem inoculation. At 24 hpi, large differences in gene expression exist in leaf and stem, and there are more differentially expressed genes and larger expression differences in leaf. The leaf is more sensitive to *S. sclerotiorum* and shows a stronger response than stem. Different defense responses appear in the leaf and stem, and the biosynthesis of lignin, callose, lectin, chitinase, PGIP, and PR protein is activated in leaf. In the stem, lipid metabolism-mediated defense responses are obviously enhanced. For the common defense responses in both leaf and stem, the chain reactions resulting from signal transduction and biological process take the primary responsibility. This research will be beneficial to exploit the potential of different tissues in plant defense and find higher resistance levels of genotypic variability in different environments. Our results are significant in the identification of resistance genes and analysis of defense mechanisms.

## 1. Introduction

*Brassica napus* (*B. napus*) is one of the most valuable crops in the world owing to its product of edible vegetable oil and livestock feed. Like other crops, oilseed rape also must tackle the threat of biotic pathogens and abiotic stress. *S. sclerotiorum* is a kind of ascomycete necrotrophic pathogen with no specific host, which can infect more than 400 different plant species, including soybean (*Glycine max*), sunflower (*Hellianthus annuus*), oilseed rape (*Brassica napus*), chickpea (*Cicer arietinum*), and even rice (*Oryza sativa* L) [1,2]. SSR caused by *S. sclerotiorum* is one of the most devastating crop diseases worldwide [1,3]. SSR causes serious crop losses around the world especially in Australia, China, Europe, and North America [4]. For instance, in China, yield losses of oilseed rape caused by SSR generally range from 10 to 20%, but may reach to 80% when it meets SSR outbreaks seasons [5,6]. In the United States, oilseed rape annual losses caused by SSR have exceeded $200 million [1]. In Australia, SSR causes yield loss of up to 20% in oilseed rape [7]. Moreover, SSR may indirectly lead to oil content and seed quality significantly declining [8,9]. In conclusion, SSR seriously impedes the development of oilseed rape; genetic improvement for oilseed rape breeding seems to be especially important.

Sclerotia of *S. sclerotiorum* can germinate myceliogenically, and ascospores, which give rise to direct infection of stem base through mycelium in the soil and aerial infection from airborne ascospores, infect plants through weak or injured plant tissues, petals, and lower stems [10,11]. Plants possess specific signaling pathways and molecular mechanisms to maximally balance growth-defense tradeoffs for plant survival. To prevent pathogen invasion, plants have evolved specific and nonspecific immune strategy profiles. In the initial stage of pathogen–host interaction, pathogen-associated molecular patterns (PAMPs) from pathogen are recognized by host transmembrane pattern recognition receptors (PRRs), then PAMPs-triggered immunity (PTI) is activated [12]. Following PTI activation, a series of sophisticated signaling pathways transmit from the cytomembrane to the nucleus through signal transduction [13,14]. The signal transduction in host plants is initiated via mitogen-activated protein kinase (MAPK) cascades, Ca^2+^ channel activation, reactive oxide species (ROS) bursts, and nitric oxide signaling, which is activated by pathogens [15,16,17]. During signal transduction, WRKY and other transcription factors activate corresponding target genes to regulate plant defense bioprocesses, such as glucosinolate, polygalacturonase inhibitor protein, polygalacturonase biosynthesis, phytoalexins, and lignin accumulation [18,19,20,21,22,23]. Simultaneously, plant hormones, jasmonic acid (JA), salicylic acid (SA), and ethylene (ET) also take participate in the regulation of defense response, they activate downstream defense genes to produce or secrete antimicrobial compounds and their derivatives [24,25,26]. In addition, some defense-related proteins and lignin, callose, also play important roles in resisting the invasion of pathogens. Lignin provides protection for various biotic and abiotic stresses like wounding or pathogen infection [27,28,29]. PGIP and callose are widely existing in a variety of higher plant cell walls, they are involved in plant development as well as multiple biotic and abiotic stresses [30]. Other proteins like chitinase and PR are involved in the degradation of components of fungal cell walls. Some other proteins like PDF1.2 and secretory protein lectin are also involved in plant defense [31,32]. These defense-related proteins play an important role in the growth of plants. Lipids are important components of membrane systems; they not only provide a physical barrier to protect plants from environmental assaults, but also energy and second messengers to influence plant development and defense [33,34,35]. Lipid metabolism is a very complicated process that includes pathogenesis and resistance mechanisms in plant–microbe interactions [36,37,38,39]. During the lipid metabolites, the hydroperoxy fatty acids are associated with the synthesis of antifungal compounds including divinyl and some plant-specific volatiles [36,40,41,42,43]. The ketodienoic fatty acids and hydroxy fatty acids accumulate during plant defense responses when attacked by pathogens [35,36]. The ketodienoic fatty acids can induce the expression of *GST1* gene and also influence cell death and ROS [44,45]. Hydroxyl acids are involved in cuticle biosynthesis, which acts as the barrier to separate from pathogens [46,47,48]. The 12-oxophytodienoic acid can be usually converted into JA and other related cyclic oxylipins, which are important signal metabolites for plant defense [49]. PTI is a primary and nonspecific defense response [16,50]. When pathogens successfully survive from PTI, they deliver avirulence effectors, which can be recognized by host disease-resistance (R) proteins, activating effector-triggered immunity (ETI), resulting in hypersensitive cell death (HR) [12,16]. Host plants resist pathogens via PTI and ETI, but PTI immune responses take responsibility for necrotrophic pathogens [51].

With the development of biotechnology and bioinformatics analysis, quantitative trait locus (QTLs) and significant genome blocks that are related to specific traits are easy to be detected, and dynamic changes of genes involved in host–pathogen interaction are easily revealed. Transcriptome analysis, bulked segregation analysis (BSA) sequencing technology, whole genome resequencing technology, and genome-wide association studies (GWAS) analysis are widely used to identify resistance-related genes [21,52,53,54,55]. Especially for RNA-seq technology, a high-efficiency technology is used for identification and expression pattern analysis of functional genes. Using RNA sequencing technology, more and more defense-related genes can be identified based on the comparative expression level and public database predictive analysis. It is a recognized auxiliary approach to selecting candidate genes. These biological technologies will also provide novel insight into potential molecular and genetic mechanisms in plant defense response to *S. sclerotiorum*. Previous research identified thousands of genes that are differently expressed in *B. napus* cv R-line compared with S-line using stem tissue infecting, and provided insight into defense against *S. sclerotiorum* [21]. Girard et al. identified an ethylene response factors family that may confer host resistance to *S. sclerotiorum* via activating genes involved in fungal recognition, redox homeostasis, and subcellular organization [53]. Wei et al. identified 17 significant associated areas and 24 genes related to SSR resistance in *B. napus*, including a glutathione S-transferase (GSTU) gene cluster [52].

No resource has been immune to SSR until now; this may be because of a lack of genotypes with resistance to the disease. Currently, only several reference genotypes are characterized as partially resistant. To identify resistant candidates under such unfavorable disadvantages, more and more attention is being paid to the resistance of different plant tissues. In this study, we analyzed the tissue-specific defense response against *S. sclerotiorum* at 24 hpi in *B. napus* based on the published transcriptome data for respective leaf and stem inoculation from National Center for Biotechnology Information (NCBI). As a result, at 24 hpi, a large difference in gene expression existed in the leaf and stem, and more genes were activated in the leaf. Different defense responses also existed in the leaf and stem; the lignin accumulation, callose deposition, and defense-related protein biosynthesis were activated in the leaf. However, lipid metabolism-mediated defense response was obviously enhanced in the stem. The signal transduction and biological process were common approaches in both leaf and stem. This research would be beneficial to exploit the potential of different tissues in plant defense and to find higher resistance levels of genotypic variability under different environments. Our results gave a better understanding of dynamic changes in SSR resistance and signaling networks in stem and leaf and were essential for varieties screening and effective breeding in oilseed rape.

## 2. Results

### 2.1. Leaf Is More Sensitive to S. sclerotiorum than Stem

There were two independent sets of transcriptome data here for the leaf and stem inoculated with *S. sclerotiorum,* respectively. Both stem inoculation (SI) experiment and leaf inoculation (LI) experiment included resistant line (R-line) and susceptible line (S-line) [21,53]. By using the FPKM (fragments per kilobase of transcript mapped onto per million fragments) value in transcriptome data, it is easy to obtain the expression level of the majority of genes. Here, we define one type of regulation expressed gene (REG) by using FPKM fold change of inoculated and mock-inoculated, if the log_2_ fold change (log_2_FC) of a gene is not equal to 0, the gene is considered as REG. REG includes up-regulation expressed gene (up-REG) and down-regulation expressed gene (down-REG), and the log_2_ FC ≥ 1 or ≤−1 represents up-REG and down-REG respectively.

According to the gene expression feedback in the two independent inoculation experiments, the leaf was more sensitive to *S. sclerotiorum* than the stem. In the SI experiment, there were a total of 8681 up-REGs and down-REGs in the R-line and S-line (Figure 1A); 152 up-REGs and 780 down-REGs were detected in the R-line, and 4386 up-REGs and 3639 down-REGs were detected in the S-line (some genes were up-REG in the R-line but were down-REG in the S-line, so the number were not directly added together) (Figure 1A; Appendix A). In the LI experiment, 44,460 REGs were detected in the R-line (ZY821) and S-line (Westar) (Figure 1A), in which 17,266 up-REGs and 19,270 down-REGs were detected in the R-line; 12,882 up-REGs and 17,412 down-REGs were detected in S-line (Westar) (Figure 1A; Appendix A). The REGs number in the LI experiment appeared higher than the SI experiment. During the SI and LI experiments, both the R-line and S-line shared many up-REGs and down-REGs. Because defense responses exist in both the R-line and S-line, these shared genes may play important roles in the R-line and S-line. For up-REGs, S-line produced 4,386 up-REGs more than 152 of R-line in SI. However, R-line produced more up-REGs in LI, and the total number of up-REGs in LI is much larger than that in SL (Figure 1B). After statistical analysis, 28 genes had up-regulated in all four lines, and 47 genes had up-regulated in two R-lines (Figure 1B). These 47 genes were speculated positively to regulate defense response. Compared with up-REGs, the down-REGs number was more in both the lines of LI and R-line of SL; this indicated that negative regulation was stronger than positive regulation (Figure 1C). Thirty-four genes down-regulated in four lines and 213 genes down-regulated in both two R-lines (Figure 1C). The 47 up-REGs and 213 down-REGs in two R-lines deserved to be further studied.

### 2.2. Leaf Expresses Stronger Defensive Response than Stem

The REGs between R-line and S-line in SI and LI experiments were then analyzed. According to the value of the log_2_FC frequency distribution, the log_2_FC value directly demonstrated the gene expression difference between SI and LI, or between S-line and R-line in corresponding SI and LI experiments. In the SI experiment, gene log_2_FC ranged from −6.77 to 6.12 in the S-line, however, it ranged from −5.32 to 3.11 in the R-line, much smaller than that of the S-line (Figure 2A,B), but the number of REGs in the S-line was far more than that of the R-line in each corresponding frequency (Figure 2A). Strangely in SI, there was only one peak value in the R-line between log_2_FC −2 to −1, but two peaks appeared in S-line with the log_2_FC −2 to −1 and 1 to 2 respectively (Figure 2A,B). In the SI experiment, there were 10,507 REGs with an average log_2_ FC value of 0.1 in the S-line. In the R-line, there were 973 REGs, and the average log_2_FC value was −0.84 (Figure 2C). However, in the LI experiment, the defensive response was stronger; the log_2_FC value ranged from −11.39 to 11.96 in the S-line and −11.10 to 10.81 in the R-line (Figure 2E). Contrary to the SI experiment, the number of REGs in the R-line was more than that of the S-line in each corresponding frequency (Figure 2D), and both peaks were located in frequencies of −2 to −1 and 1 to 2 in the S-line and R-line, respectively (Figure 2D,E). There were 33,441 REGs, and the average log_2_FC value was −0.18 in the S-line; however, in the R-line, there were 40,027 REGs with an average log_2_FC value of −0.13 (Figure 2F). The amplitude of log_2_FC was bigger in LI than in the SI experiment, so the leaf expressed a stronger defensive response to *S. sclerotiorum* than the stem.

### 2.3. Biosynthesis of Lignin and Defense-Related Compounds Are Activated in Leaf

Lignin is one kind of important secondary metabolite in terrestrial plants; lignin is polymerized by heterogeneous monolignols at the surface of the secondary cell wall. Lignin also provides protection for various biotic and abiotic stresses, like wounding or pathogen infection [27,28,29]. By using genetic modification, lignin content and composition can be controlled by altering single or multiple lignin biosynthesis gene expressions, which leads to corresponding phenotypic trait variation [56]. In this research, there were 17,219 up-REGs only presented in the R-line of the LI experiment (Appendix A), and we found that the lignin biosynthesis catalytic enzymes PAL, C4H, 4CL, HCT, CCR, CSE, COMT, CCoAOMT, F5H, CAD, and LAC were included and up-regulated (Figure 3A). Especially PAL2, CAD5, and CAD8, in which all the corresponding homologous genes had up-regulated, and 9 out of 10 C4H genes were up-regulated. There were no related up-REGs detected expression in the SI experiment at 24 hpi. Obviously, lignin biosynthesis was active in the leaf after *S. sclerotiorum* inoculation (Figure 3A). Additionally, defense-related proteins played an important role in the growth of plants, some of them were also induced to express, including PR proteins, lectin, callose, chitinase, PGIP, and PDF family (Figure 3B). The results showed that the biosynthesis of lignin and defense-related proteins were active in leaf defense response.

To verify the transcriptome data of the LI experiment, the leaves of the R-line and S-line were inoculated with *S. sclerotiorum*, the R-line showed less necrosis at 36 hpi (Figure 4A). The RNA of 0 hpi, 24 hpi, and 36 hpi were extracted for further research. The qRT-PCR was performed to verify the difference between 0 hpi and 24 hpi. Four genes were selected to detect the change of expression level in the R-line. As a result, the expression was increased at 24 hpi and showed a significant difference with 0 hpi; the results of qRT-PCR experiments were consistent with the transcriptome results (Figure 4B). As shown in Figure 3, The lignin synthase CAD5 (BnaC07g44880D) and defensive protein PR4 (BnaA03g28770D) were up-regulated at 24 hpi. To further study the dynamic change of their expression level during the interaction of *S. sclerotiorum* and *B. napus*, the expression levels of CAD5 and PR4 at 0 hpi, 24 hpi, and 36 hpi were detected, and the CAD5 was up-regulated at 24 hpi, and then decreased at 36 hpi in both the R-line and S-line; the expression level was always higher in the R-line and showed a significant difference to the S-line (Figure 4C). The PR4 was up-regulated at 24 hpi and maintained the expression level until 36 hpi in the R-line. In the S-line, the expression level was slightly up-regulated at 24 hpi and 36 hpi; the expression level in R-line was significantly higher than in S-line (Figure 4C).

### 2.4. Lipid Metabolism Is Significant in Stem

Lipids are important components of membrane systems; they provide not only a physical barrier to protect plants from environmental assaults, but also energy and second messengers to influence plant development and defense [33,34,35]. Lipid metabolism is a very complicated process that includes pathogenesis and resistance mechanisms in plant–microbe interactions [36,37,38,39]. In this research, the unique 105 up-REGs that were only presented in the R-line of the SI experiment were performed with KEGG (Kyoto Encyclopedia of Genes and Genomes) enrichment analysis and pathway illustration (Appendix A). The KEGG enrichment results indicated that these genes mainly participated in lipid metabolism (Figure 5A; Appendix A). The result of pathway analysis also showed that they participated in lipid and fatty acid metabolism processes, such as sphingolipid metabolism, glycosphingolipid biosynthesis, linoleic acid metabolism, alpha-linoleic acid metabolism, etc. (Figure 5B; Appendix A). During the lipid metabolites, the hydroperoxy fatty acids, ketodienoic fatty acids, hydroxy fatty acids, and 12-oxophytodienoic acid were all involved in lipid metabolism and plant defense (Figure 5C). Additionally, the lipoxygenase synthesis genes were also up-regulated (Figure 5C). All these results suggested that lipid metabolism played an important role in plant stem defense response.

### 2.5. Signal Transduction and Biological Process Are Common Approaches in Both Leaf and Stem

There were 105 unique up-REGs in the R-line of SI and 17,219 unique up-REGs in the R-line of LI; they shared 47 genes that were up-regulated in the R-line of both SL and LI experiments (Figure 6A; Appendix A), and the expression level of some genes were verified by qRT-PCR (Appendix A). These 47 genes played important roles in the defense response of plant leaves and stems. It is curious what molecular function they had and which biological process they participated in. After Gene Ontology (GO) enrichment and classification, we found that 34 out of 47 genes participated in response to stimulus, 34 genes participated in the single-organism process, and 32 genes participated in the metabolic process (Figure 6B; Appendix A). In addition, more than 10 genes were involved in other biological processes including cellular regulation, biological regulation, cellular component organization or biogenesis, and regulation of biological processes (Figure 6B). The result indicated that most of these genes were involved in signal transduction and secondary signal molecules during plant defense response. Moreover, 24 genes have the molecular functions of catalytic activity, and 38 genes were involved in the cell part of the cellular component (Figure 6B). This result revealed that some biochemical enzymes were active in plant–microbe interactions. Furthermore, as shown in the GO-enrichment result, many genes were related to glutathione transferase activity, alkyl or aryl groups transferase activity, UDP-glucosyltransferase activity, glucosyltransferase activity, UDP-glycosyltransferase activity, and hexosyl groups transferase activity (Figure 6C). This result illustrated that the plant defense response was a methodical consecutive reaction and needed the cooperation of multiple processes.

## 3. Discussion

The plant defense response initiates from pathogen recognition, through a precise and strict regulatory network and then activates the downstream defense response [57]. Lots of genes are involved in this response process, they coordinate with each other to transcribe and translate various types of molecules to confront biotic stresses. However, with the help of high-throughput RNA sequencing technology, we can easily identify most of the functional genes based on expression and annotation [58]. The defense response is a continuous process, with specific defense responses present at different times. In this research, we analyzed the unique up-REGs in leaf and stem, and in both leaf and stem at 24 hpi. As a result, defense-related proteins were up-regulated only in the leaf at 24 hpi (Figure 3); this did not mean that defense-related proteins were not up-regulated in the stem, it was just not detected at 24 hpi. A previous study showed that defense-related proteins were up-regulated at 48 hpi and 96 hpi in stem [21]. In addition, plant hormones and glucosinolates played an important role in defense response; here we also checked changes in the expression of genes related to hormone synthesis and signaling pathways, because they were not up-regulated at 24 hpi or have no tissue specificity. In this research, the defense mechanism at 24 hpi was partially briefly represented in Figure 7.

Before infection, *S. sclerotiorum* needed to revise a feasible condition to penetrate, including access to exogenous nutrients, compound appressorium development, and balanced oxalic acid (OA) metabolism [59]. At the early infection phase, pathogens used compound appressoria to achieve cuticle layer penetration but not the underlying host epidermal cell to form penetration pegs [60]; then the penetration pegs led to the colonization of subcuticular bulbous hyphae [59,61,62]. At this time, the first phase of plant defense was initiated, and the defense-related genes were quickly expressed. Subsequently, this led to physical and chemical defense barriers including callose deposition, ROS, and glucosinolates accumulation [59,63], and the defense-related proteins were also activated [21]. These results were also consistent with our research (Figure 3B). However, if the pathogens successfully suppressed the host basal defense, they next spread filamentous subcuticular hyphae and ramified infection hyphae through the primary arsenal such as OA, OA-independent toxins, and cell wall-degrading enzymes to kill the host cell and transform to necrotrophy, subsequently absorbing nutrients [1,59]. However, at this time, the host cells would take measures to prevent the invasion of pathogens, such as expressing oxalate-degrading enzyme and PGIP [64,65]. As shown in Figure 4, PGIP and other defense proteins were up-regulated in leaf at 24 hpi. Here, we also noted that at least one of the homologous genes rather than all the homologous genes were up-regulated. In this research, defense-related proteins were up-regulated in the leaf but not in the stem at 24 hpi; this did not mean that there was no such defense in the stem, maybe the reaction appeared earlier or later.

There were no resources for absolute immunity to *S. sclerotiorum* until now, only several reference cultivar lines were characterized as partially resistant, so more and more attention was paid to SSR management like biological and chemical control according to different plant tissues. The tissue of leaf, stem, and root was investigated to achieve sufficiently significant diversity in responses to *S. sclerotiorum* infection [21,53,66]. Finding higher resistance levels to improve genotypic variability based on tissue specificity seemed to be a potential option. Unlike other tissues, root infestation was rarely observed under natural conditions, root infection was also seldomly reported. Recent research showed that synthesized compounds of primary GSL (glucoiberverin, glucoerucin, and glucoberteroin) were accumulated in the root after infection, while the subsequent compounds were more present in the stem [67]. On the contrary, the stem and leaves are the main components of the aerial parts of the oilseed rape plant, they are most likely to be infected by *S. sclerotiorum*. The disease resistance strength of the plant mainly depends on the disease resistance of leaves and stems. The tissue structure and cells of leaves and stems were different, which led to the difference in response speed and pathway to pathogenic bacteria. In this research, we illustrated the unique defensive response in stem and leaf at 24 hpi. The lignin and defense-related protein up-regulated uniquely in leaf (Figure 3), and lipid metabolism took responsibility obviously in the stem (Figure 4). Flower petals were susceptible tissue for *S. sclerotiorum*, but we did not discuss them in this research. These results provided approaches to the identification and characterization of tissue-specific properties and were beneficial to preventing SSR in agricultural praxis.

Bacteria and fungi are in constant competition with each other in the soil, some mycoparasitic fungi can destroy *S. sclerotiorum*. *Coniothyrium minitans* (*C. minitans*) is one kind of mycoparasitic fungus of *S. sclerotiorum*; *C. minitans* extracellular lytic enzymes, such as chitinase help to penetrate the host cell and facilitate the mycoparasitic to *S. sclerotiorum* [68]. *C. minitans* biological agent has been widely used in agricultural production to control diseases caused by *S. sclerotiorum*, especially in the production of rapeseed and many other vegetable crops [69]. Chitin is an important component of the pathogenic fungi cell walls; chitinase can hydrolyze chitin to inhibit fungal growth and induce the following defense response in host plants [68]. Recent research demonstrated that the overexpression of a *C. minitans* chitinase CMCH1 enhanced resistance to *S. sclerotiorum* in soybean. After inoculation, the transgenic plants showed increased accumulation of ROS and high expression of defense-related genes as well as enzymes [70]. The effect of mycoparasitic fungus on *S. sclerotiorum* provides a novel idea for plant disease-resistance improvement.

## 4. Materials and Methods

### 4.1. Plant and Fungal Pathogen Materials and Leaf Inoculation

*B. napus* cultivar line Westar and ZY821 were collected from our laboratory and were planted in our experimental field in October in Yangluo, Wuhan, Hubei province, China. They were planted in three replicates and developed by self-pollination; all the lines were purified for many years. The *S. sclerotiorum* physiological race 1980 was preserved in our laboratory; the mycelia were cultivated on potato dextrose agar medium at 24 °C in the dark. The advancing edge of growing mycelia was selected to make the mycelia plug for inoculation. Leaves were of the same phase and consistent size and were subjected to leaf inoculation using special mycelium plugs [71]. Samples with treatment were collected for RNA extraction. The *S. sclerotiorum* strain SS-1 used in the stem inoculation experiment was collected from the field and stored in National Key Laboratory of Crop Genetic Improvement, Huazhong Agricultural University, Wuhan 430070, China. The *S. sclerotiorum* strain used in leaf inoculation experiment was collected from the field and stored in Department of Biological Sciences, University of Manitoba, Winnipeg, MB R3T 2N2, Canada. The *S. sclerotiorum* strain used for RNA extraction and qRT-PCR was collected from the field and stored in The Key Laboratory of Biology and Genetic Improvement of Oil Crops, the Ministry of Agriculture and Rural Affairs of the PRC, Oil Crops Research Institute, Chinese Academy of Agricultural Sciences, Wuhan 430062, China.

### 4.2. RNA Extraction and Quantitative Real-Time PCR Analysis

The qRT-PCR experiment was performed to verify the leaf inoculation RNA-seq result. The total RNA of Westar and ZY821 was extracted using TRIzol reagent (Invitrogen, Carlsbad, CA, USA). Each sample before and after inoculation was collected from three biological replicates. Two micrograms of total RNA was used to make cDNA using the PrimeScript™ RT reagent Kit with gDNA Eraser (TaKaRa Co., Ltd., Beijing, China). The *B. napus β*-actin gene (AF111812) was used as a reference standard. The relative expression level was calculated using the 2^−∆∆Ct^ method [72].

### 4.3. Transcriptome Data Sources and Processing

This research was conducted based on published data [21,53]. The LI transcriptome data (GSE81545) and SI transcriptome data (accession no. SRP053361) were obtained from NCBI (https://www.ncbi.nlm.nih.gov/, accessed on 30 March 2021). The FPKM was obtained from the corresponding article.

### 4.4. KEGG Enrichment, KEGG Pathway Analysis and GO Enrichment Analysis

The KEGG enrichment analysis was performed through TBtools [73], and the KEGG pathway analysis and GO enrichment analysis were performed by using the OmicShare tools, (https://www.omicshare.com/tools/, accessed on 21 February 2022), a free online platform for data analysis.

## 5. Conclusions

In this research, transcriptome analysis revealed the defense responses to *S. sclerotiorum* in the leaf and stem at 24 hpi. A large difference in gene expression existed in the leaf and stem; the leaf was more sensitive to *S. sclerotiorum* and showed a stronger response than the stem. There were more genes up-regulated and down-regulated, and the genes had a greater fold change in the leaf. After inoculation with *S. sclerotiorum*, the biosynthesis of lignin, callose, lectin, chitinase, PGIP, and PR protein was uniquely activated in the leaf. However, uniquely in the stem, lipid metabolism-mediated defense response was intensive and was obviously enhanced. For the common defense responses in both leaf and stem, the chain reaction resulted from signal transduction and the biological process assumed the primary responsibility. Our results were significant in the identification of resistance genes and analysis of defense mechanisms.

## Figures and Tables

**Figure 1 plants-11-02001-f001:**
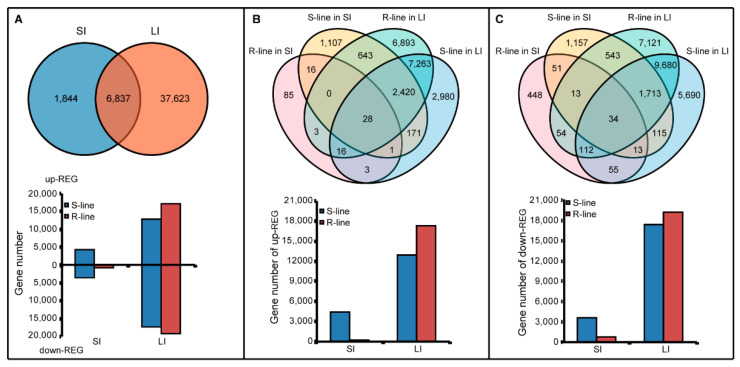
Statistical analysis for numbers of up-REGs and down-REGs. (**A**) Venn diagram and histogram for up-REGs and down-REGs number of R-line and S-line in SI and LI experiments. (**B**) Venn diagram and histogram for the up-REGs number of R-line and S-line in SI and LI experiments. (**C**) Venn diagram and histogram for the down-REGs number of R-line and S-line in SI and LI experiments.

**Figure 2 plants-11-02001-f002:**
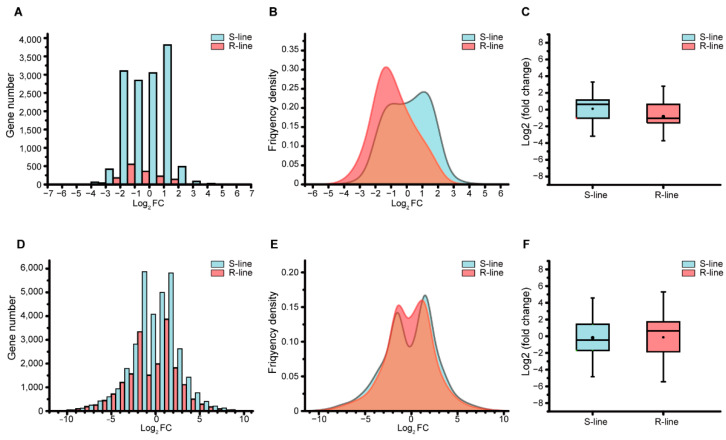
Gene number, frequency distribution, and statistics difference of log*_2_*FC value in SI and LI. (**A**) Corresponding gene number of log_2_FC in R-line and S-line of SI experiment. (**B**) Corresponding frequency distribution of log_2_FC in R-line and S-line of SI experiment. (**C**) Statistics difference of log_2_FC value in R-line and S-line of SI experiment. (**D**) Corresponding gene number of log_2_FC in R-line and S-line of LI experiment. (**E**) Corresponding frequency distribution of log_2_FC in R-line and S-line of LI experiment. (**F**) Statistics difference of log_2_FC value in R-line and S-line of LI experiment.

**Figure 3 plants-11-02001-f003:**
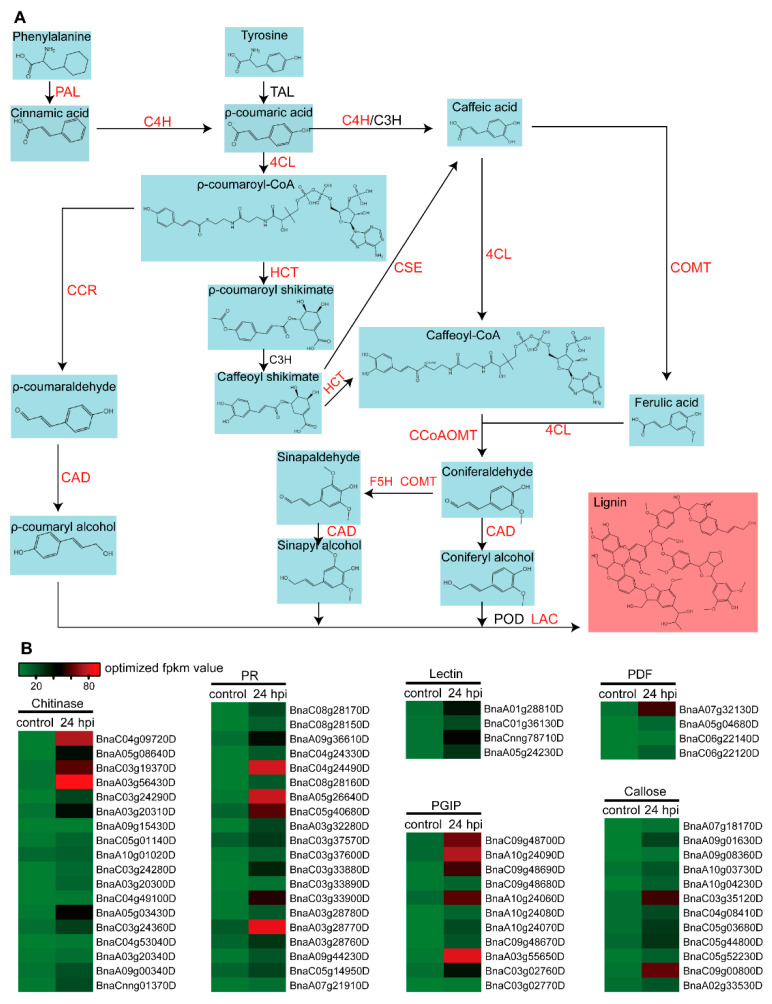
Lignin biosynthesis pathway and changes in expression of defense-related genes. (**A**) Lignin biosynthesis pathway; words in red color mean the gene was up-regulated. PAL: phenylalanine ammonia-lyase; C4H: cinnamate 4-hydroxylase; TAL: tyrosine ammonia-lyase; C3H: p-coumarate 3-hydroxylase; 4CL: 4-coumarate CoA ligase; CCR: cinnamoyl-CoA reductase; HTC: Hydroxycinnamoyl-CoA transferase (HCT); CCoAOMT: caffeoyl-CoA O-methyltransferase; F5H: ferulate 5-hydroxylase; CSE: caffeoyl shikimate esterase; COMT: caffeic acid O-methyltransferase; CAD: cinnamyl alcohol dehydrogenase; LAC; laccase; POD: peroxidase. (**B**) Expression heat map of defense-related genes before and after inoculation based on FPKM value.

**Figure 4 plants-11-02001-f004:**
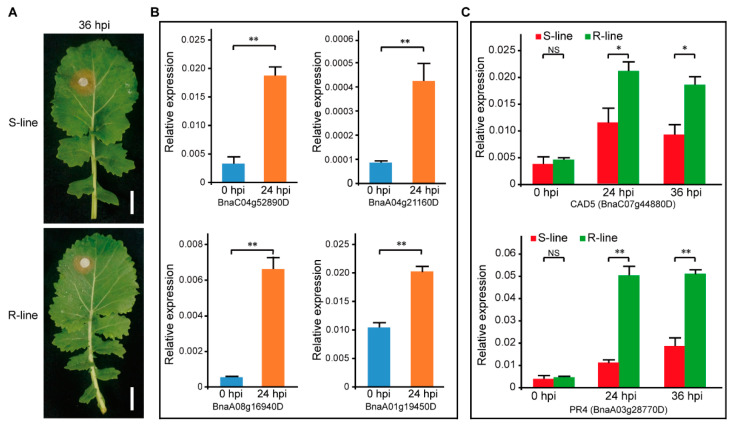
Leaf inoculation of Westar (S-line), ZY821 (R-line), and qRT-PCR experiments. (**A**) Phenotype photos of Westar and ZY821 at 36 hpi, bar = 2 cm. (**B**) The expression level of four up-REGs in R-line at 0 hpi and 24 hpi. (**C**) Dynamic change of the expression level of CAD5 and PR4 in R-line and S-line during the interaction of *S. sclerotiorum* and *B. napus*. ‘*’ and ‘**’ means *p* < 0.05 and *p* < 0.01 of *t*-test.

**Figure 5 plants-11-02001-f005:**
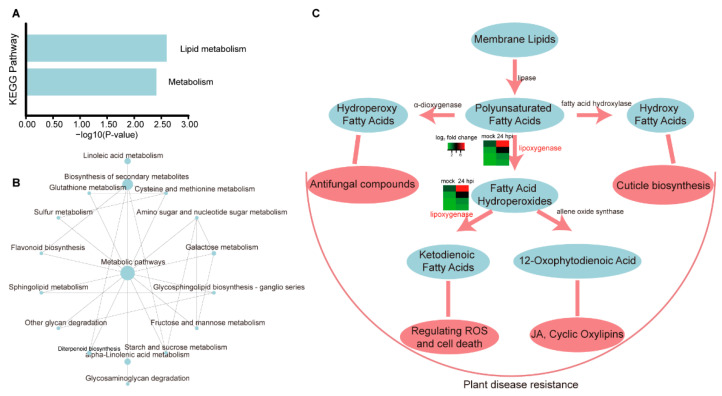
KEGG enrichment, pathway analysis, and schematic diagram of lipid metabolism-mediated defense response. (**A**) KEGG enrichment analysis of 105 up-REGs in SI experiment. (**B**) Putative KEGG pathway model of 105 up-REGs in SI experiment. (**C**) Schematic diagram of lipid metabolism in disease resistance. Words in red color mean gene up-regulation is expressed.

**Figure 6 plants-11-02001-f006:**
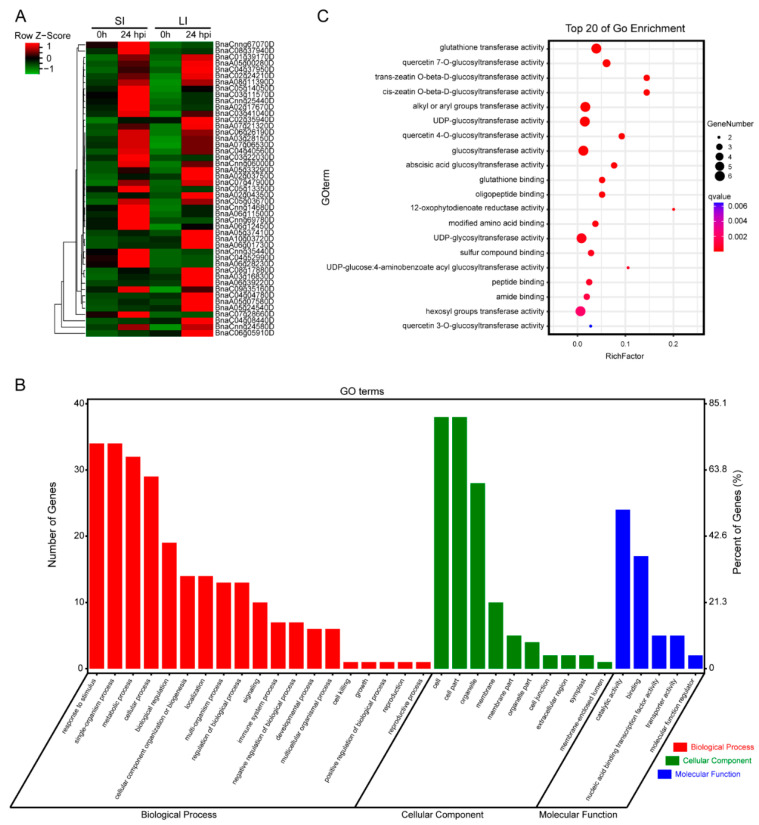
Expression heat map and GO enrichment of 47 shared up-REGs. (**A**) Expression heat map of the 47 up-REGs shared in SI and LI experiment based on FPKM value. (**B**) GO enrichment is involved in biological processes and cellular and molecular functions. (**C**) Top 20 GO classifications of the 47 up-REGs shared in SI and LI experiment.

**Figure 7 plants-11-02001-f007:**
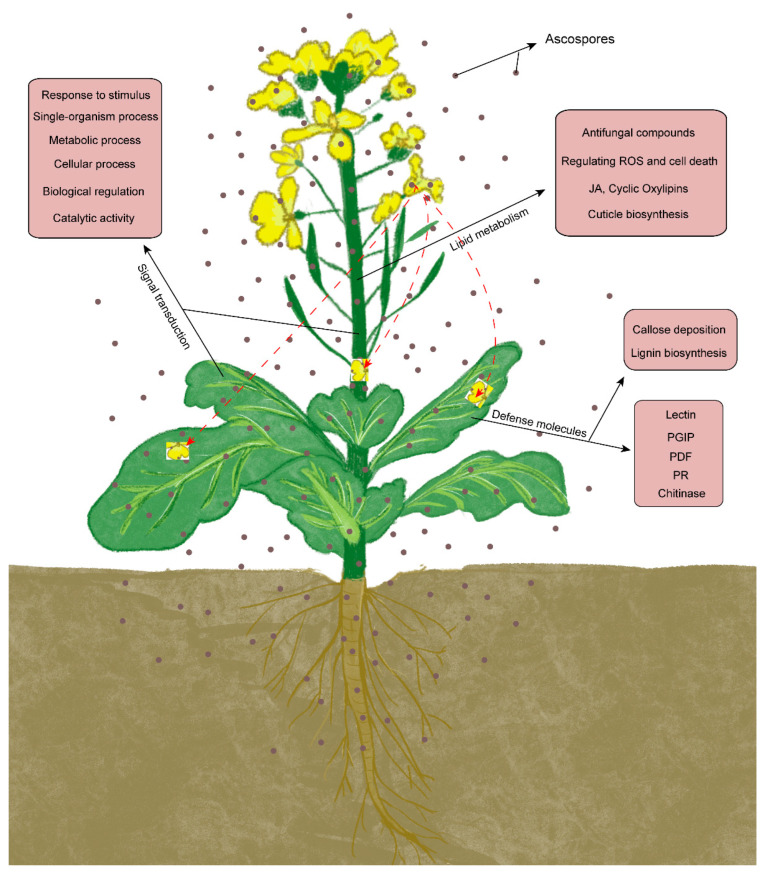
A brief summary of the defense mechanisms of oilseed rape against *S. sclerotiorum* at 24 hpi.

## Data Availability

The referenced contributions can be obtained from corresponding articles. Some of the original data in this study are represented in the Appendix A.

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
