# Peer review of "Analysis of Tissue-Specific Defense Responses to Sclerotinia sclerotiorum in Brassica napus"

_plants, 2022, doi:10.3390/plants11152001_

Round 1

Reviewer 1 Report

1. This study detected the expression patterns of plant tissues when infected with Sclerotinia sclerotiorum which causes Sclerotinia stem rot in Brassica napus and identified significant tissue-specific defense responses, the weak point mainly the lack of statistic analysis in the experimental design. The authors need to conduct qPCR experiments at least for three replicates and then perform an ANOVA test for providing the results of significant differences between different tissues of expression patterns. Additionally, the authors need to select some critical key genes to further analyze their expression in the interaction of Sclerotinia sclerotiorum and Brassica napus at critical stages.

2. The minor point is the grammar errors throughout the manuscript, several examples are listed as follows.

2.1. has revealed large number - has revealed a large number

2.2. large difference in gene expression exist in leaf and stem - large differences in gene expression exist in leaf and stem

2.3. Leaf is more sensitive to S. sclerotiorum, and shows stronger response - The leaf is more sensitive to S. sclerotiorum, and shows a stronger response

2.4. the chain reactions resulted - the chain reactions resulting

2.5. genotypic variability in different environment - genotypic variability in different environments

2.6. are significant in identification - are significant in the identification

2.7. The authors need to check grammar throughout the manuscript.

3. The layout of the text in figure 6 should be unified.

4. M&M: Need to add a part of statistical analysis.

5. Iconic photos of the control plants as well as the infected plants should be added to the main manuscript.

Author Response

Point 1: This study detected the expression patterns of plant tissues when infected with Sclerotinia sclerotiorum which causes Sclerotinia stem rot in Brassica napus and identified significant tissue-specific defense responses, the weak point mainly the lack of statistic analysis in the experimental design. The authors need to conduct qPCR experiments at least for three replicates and then perform an ANOVA test for providing the results of significant differences between different tissues of expression patterns. Additionally, the authors need to select some critical key genes to further analyze their expression in the interaction of Sclerotinia sclerotiorum and Brassica napus at critical stages.

Response 1: Thanks, we have performed the qRT-PCR experiment for some up-regulated genes in leaf inoculation. During the experiment, each sample had three biological repetitions, and we performed the ANOVA test, the results were consistent with transcriptome results. We also have selected two PR genes and detected the dynamic change of their expression level during the interaction of Sclerotinia sclerotiorum and Brassica napus. We have added the results to the new manuscript. Because we did not get the corresponding rapeseed varieties of stem inoculation, therefore we had not performed the qRT-PCR experiment of stem samples.

Point 2: The minor point is the grammar errors throughout the manuscript, several examples are listed as follows.

Response 2: Thanks, we have the manuscript checked by an online tool, and corrected the grammar errors.

Point 3: The layout of the text in figure 6 should be unified.

Response 3: Yes, we have now corrected.

Point 4: M&M: Need to add a part of statistical analysis.

Response 4: We have added some qRT-PCR experiments, and performed the ANOVA test.

Point 5: Iconic photos of the control plants as well as the infected plants should be added to the main manuscript.

Response 5: Yes, we repeated the leaf inoculation experiment and now added the Iconic photos of leaf inoculation. We did no add the Iconic photos of stem inoculation due to lack of corresponding rapeseed varieties, the corresponding article showed the photos.

Reviewer 2 Report

The manuscript "Analysis of tissue-specific defense responses to Sclerotinia sclerotiorum in Brassica napus" by Liu et al. presents original research results.

I believe that the research was implemented correctly based on appropriate research methods. The manuscript was prepared very carefully therefore I recommend publication of the work after making small corrections.

1 Please clarify at what phase the infection was carried out?

2 Please state in which collection the S. sclerotiorum strain is stored. Give the specific name of the laboratory.

3. On what equipment was the qRT-PCR study performed?

4. Supplementary materials are not included in the evaluation of the manuscript requires correction.

Author Response

Point 1: Please clarify at what phase the infection was carried out?

Response 1: In the Leaf inoculation experiment, the infection was carried at 30-50% bloom stage. The stem inoculation experiment, the infection was carried out at the termination of flowering. We have now added in the manuscript.

Point 2: Please state in which collection the S. sclerotiorum strain is stored. Give the specific name of the laboratory.

Response 2: The S. sclerotiorum strain SS-1 used in stem inoculation experiment was collected from field, and stored in National Key Laboratory of Crop Genetic Improvement, Huazhong Agricultural University, Wuhan 430070, China. The S. sclerotiorum strain used in leaf inoculation experiment was collected from field, and stored in Department of Biological Sciences, University of Manitoba, Winnipeg, MB R3T 2N2, Canada. The S. sclerotiorum strain used for RNA extraction and qRT-PCR was collected from field, and stored in The Key Laboratory of Biology and Genetic Improvement of Oil Crops, the Ministry of Agriculture and Rural Affairs of the PRC, Oil Crops Research Institute, Chinese Academy of Agricultural Sciences, Wuhan 430062, P.R. China. The S. sclerotiorum strain had no race differences. We have now added in the manuscript.

Point 3: On what equipment was the qRT-PCR study performed?

Response 3: The qRT-PCR study was performed through CFX ConnectTM Real-Time System. We have now added in the manuscript.

Point 4: Supplementary materials are not included in the evaluation of the manuscript requires correction.

Response 4: Thanks, We have now corrected in the manuscript.

Round 2

Reviewer 1 Report

The revised version looks better.